# MobileViT: Light-weight, General-purpose, and Mobile-friendly Vision Transformer

**Sachin Mehta**
Apple

**Mohammad Rastegari**
Apple

## Abstract

Light-weight convolutional neural networks (CNNs) are the de-facto for mobile vision tasks. Their spatial inductive biases allow them to learn representations with fewer parameters across different vision tasks. However, these networks are spatially local. To learn global representations, self-attention-based vision transformers (ViTs) have been adopted. Unlike CNNs, ViTs are heavyweight. In this paper, we ask the following question: *is it possible to combine the strengths of CNNs and ViTs to build a light-weight and low latency network for mobile vision tasks?* Towards this end, we introduce MobileViT, a light-weight and general-purpose vision transformer for mobile devices. MobileViT presents a different perspective for the global processing of information with transformers. Our results show that MobileViT significantly outperforms CNN- and ViT-based networks across different tasks and datasets. On the ImageNet-1k dataset, MobileViT achieves top-1 accuracy of 78.4% with about 6 million parameters, which is 3.2% and 6.2% more accurate than MobileNetv3 (CNN-based) and DeIT (ViT-based) for a similar number of parameters. On the MS-COCO object detection task, MobileViT is 5.7% more accurate than MobileNetv3 for a similar number of parameters. Our source code is open-source and available at: https://github.com/apple/ml-cvnets.

## 1 Introduction

Self-attention-based models, especially vision transformers (ViTs; Figure 1a; Dosovitskiy et al., 2021), are an alternative to convolutional neural networks (CNNs) to learn visual representations. Briefly, ViT divides an image into a sequence of non-overlapping patches and then learns inter-patch representations using multi-headed self-attention in transformers (Vaswani et al., 2017). The general trend is to increase the number of parameters in ViT networks to improve the performance (e.g., Touvron et al., 2021a; Graham et al., 2021; Wu et al., 2021). However, these performance improvements come at the cost of model size (network parameters) and latency. Many real-world applications (e.g., augmented reality and autonomous wheelchairs) require visual recognition tasks (e.g., object detection and semantic segmentation) to run on resource-constrained mobile devices in a timely fashion. To be effective, ViT models for such tasks should be light-weight and fast. Even if the model size of ViT models is reduced to match the resource constraints of mobile devices, their performance is significantly worse than light-weight CNNs. For instance, for a parameter budget of about 5-6 million, DeIT (Touvron et al., 2021a) is 3% less accurate than MobileNetv3 (Howard et al., 2019). Therefore, the need to design light-weight ViT models is imperative.

Light-weight CNNs have powered many mobile vision tasks. However, ViT-based networks are still far from being used on such devices. Unlike light-weight CNNs that are easy to optimize and integrate with task-specific networks, ViTs are heavy-weight (e.g., ViT-B/16 vs. MobileNetv3: 86 vs. 7.5 million parameters), harder to optimize (Xiao et al., 2021), need extensive data augmentation and L2 regularization to prevent over-fitting (Touvron et al., 2021a; Wang et al., 2021), and require expensive decoders for down-stream tasks, especially for dense prediction tasks. For instance, a ViT-based segmentation network (Ranftl et al., 2021) learns about 345 million parameters and achieves similar performance as the CNN-based network, DeepLabv3 (Chen et al., 2017), with 59 million parameters. The need for more parameters in ViT-based models is likely because they lack image-specific inductive bias, which is inherent in CNNs (Xiao et al., 2021). To build robust and high-performing ViT models, hybrid approaches that combine convolutions and transformers

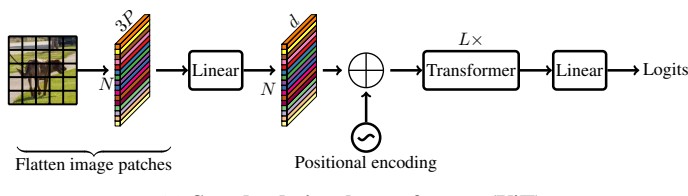

(a) **Standard visual transformer (ViT)**

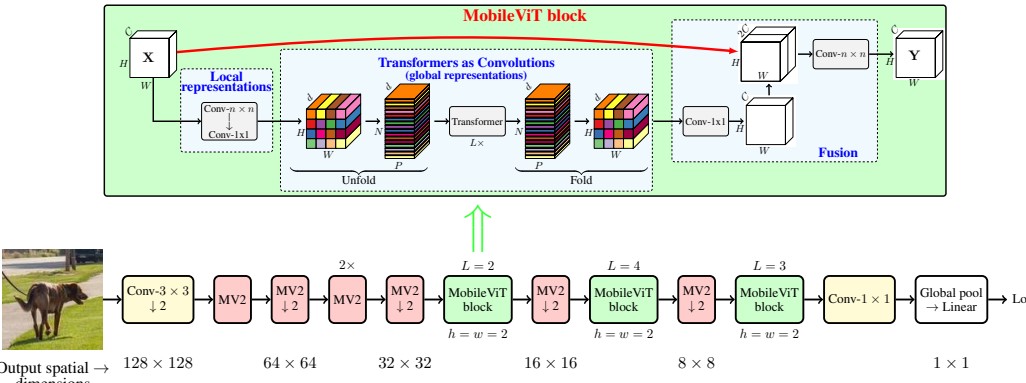

(b) **MobileViT**. Here, Conv-$n \times n$ in the MobileViT block represents a standard $n \times n$ convolution and MV2 refers to MobileNetv2 block. Blocks that perform down-sampling are marked with $\downarrow 2$.

Figure 1: **Visual transformers vs. MobileViT**

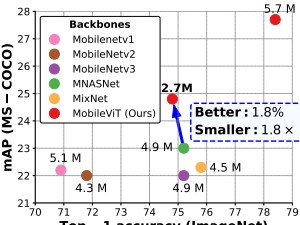

Figure 2: **MobileViT shows better task-level generalization properties** as compared to light-weight CNN models. The network parameters are listed for SSDLite network with different feature extractors (MobileNetv1 (Howard et al., 2017), MobileNetv2 (Sandler et al., 2018), MobileNetv3 (Howard et al., 2019), MNASNet (Tan et al., 2019), MixNet (Tan & Le, 2019b), and MobileViT (Ours)) on the MS-COCO dataset.

are gaining interest (Xiao et al., 2021; d'Ascoli et al., 2021; Chen et al., 2021b). However, these hybrid models are still heavy-weight and are sensitive to data augmentation. For example, removing CutMix (Zhong et al., 2020) and DeIT-style (Touvron et al., 2021a) data augmentation causes a significant drop in ImageNet accuracy (78.1% to 72.4%) of Heo et al. (2021).

It remains an open question to combine the strengths of CNNs and transformers to build ViT models for mobile vision tasks. Mobile vision tasks require light-weight, low latency, and accurate models that satisfy the device's resource constraints, and are general-purpose so that they can be applied to different tasks (e.g., segmentation and detection). Note that floating-point operations (FLOPs) are not sufficient for low latency on mobile devices because FLOPs ignore several important inference-related factors such as memory access, degree of parallelism, and platform characteristics (Ma et al., 2018). For example, the ViT-based method of Heo et al. (2021), PiT, has 3× fewer FLOPs than DeIT (Touvron et al., 2021a) but has a similar inference speed on a mobile device (DeIT vs. PiT on iPhone-12: 10.99 ms vs. 10.56 ms). Therefore, instead of optimizing for FLOPs[1], this paper focuses on designing a **light-weight** (§3), **general-purpose** (§4.1 & §4.2), and **low latency** (§4.3) network for mobile vision tasks. We achieve this goal with MobileViT that combines the benefits of CNNs (e.g., spatial inductive biases and less sensitivity to data augmentation) and ViTs (e.g., input-adaptive weighting and global processing). Specifically, we introduce the MobileViT block that encodes both local and global information in a tensor effectively (Figure 1b). Unlike ViT and its variants (with and without convolutions), MobileViT presents a different perspective to learn global representations. Standard convolution involves three operations: unfolding, local processing, and

---

[1]MobileViT FLOPs can be further reduced using existing methods (e.g., DynamicViT (Rao et al., 2021)).

folding. MobileViT block replaces local processing in convolutions with global processing using transformers. This allows MobileViT block to have CNN- and ViT-like properties, which helps it learn better representations with fewer parameters and simple training recipes (e.g., basic augmentation). To the best of our knowledge, this is the first work that shows that light-weight ViTs can achieve light-weight CNN-level performance with simple training recipes across different mobile vision tasks. For a parameter budget of about 5-6 million, MobileViT achieves a top-1 accuracy of 78.4% on the ImageNet-1k dataset (Russakovsky et al., 2015), which is 3.2% more accurate than MobileNetv3 and has a simple training recipe (MobileViT vs. MobileNetv3: 300 vs. 600 epochs; 1024 vs. 4096 batch size). We also observe significant gains in performance when MobileViT is used as a feature backbone in highly optimized mobile vision task-specific architectures. Replacing MNASNet (Tan et al., 2019) with MobileViT as a feature backbone in SSDLite (Sandler et al., 2018) resulted in a better (+1.8% mAP) and smaller ($1.8\times$) detection network (Figure 2).

## 2 RELATED WORK

**Light-weight CNNs.** The basic building layer in CNNs is a standard convolutional layer. Because this layer is computationally expensive, several factorization-based methods have been proposed to make it light-weight and mobile-friendly (e.g., Jin et al., 2014; Chollet, 2017; Mehta et al., 2020). Of these, separable convolutions of Chollet (2017) have gained interest, and are widely used across state-of-the-art light-weight CNNs for mobile vision tasks, including MobileNets (Howard et al., 2017; Sandler et al., 2018; Howard et al., 2019), ShuffleNetv2 (Ma et al., 2018), ESPNetv2 (Mehta et al., 2019), MixNet (Tan & Le, 2019b), and MNASNet (Tan et al., 2019). These light-weight CNNs are versatile and easy to train. For example, these networks can easily replace the heavy-weight backbones (e.g., ResNet (He et al., 2016)) in existing task-specific models (e.g., DeepLabv3) to reduce the network size and improve latency. Despite these benefits, one major drawback of these methods is that they are spatially local. This work views transformers as convolutions; allowing to leverage the merits of both convolutions (e.g., versatile and simple training) and transformers (e.g., global processing) to build light-weight (§3) and general-purpose (§4.1 and §4.2) ViTs.

**Vision transformers.** Dosovitskiy et al. (2021) apply transformers of Vaswani et al. (2017) for large-scale image recognition and showed that with extremely large-scale datasets (e.g., JFT-300M), ViTs can achieve CNN-level accuracy without image-specific inductive bias. With extensive data augmentation, heavy L2 regularization, and distillation, ViTs can be trained on the ImageNet dataset to achieve CNN-level performance (Touvron et al., 2021a;b; Zhou et al., 2021). However, unlike CNNs, ViTs show substandard optimizability and are difficult to train. Subsequent works (e.g., Graham et al., 2021; Dai et al., 2021; Liu et al., 2021; Wang et al., 2021; Yuan et al., 2021b; Chen et al., 2021b) shows that this substandard optimizability is due to the lack of spatial inductive biases in ViTs. Incorporating such biases using convolutions in ViTs improves their stability and performance. Different designs have been explored to reap the benefits of convolutions and transformers. For instance, ViT-C of Xiao et al. (2021) adds an early convolutional stem to ViT. CvT (Wu et al., 2021) modifies the multi-head attention in transformers and uses depth-wise separable convolutions instead of linear projections. BoTNet (Srinivas et al., 2021) replaces the standard $3\times3$ convolution in the bottleneck unit of ResNet with multi-head attention. ConViT (d'Ascoli et al., 2021) incorporates soft convolutional inductive biases using a gated positional self-attention. PiT (Heo et al., 2021) extends ViT with depth-wise convolution-based pooling layer. Though these models can achieve competitive performance to CNNs with extensive augmentation, the majority of these models are heavy-weight. For instance, PiT and CvT learns $6.1\times$ and $1.7\times$ more parameters than EfficientNet (Tan & Le, 2019a) and achieves similar performance (top-1 accuracy of about 81.6%) on ImageNet-1k dataset, respectively. Also, when these models are scaled down to build light-weight ViT models, their performance is significantly worse than light-weight CNNs. For a parameter budget of about 6 million, ImageNet-1k accuracy of PiT is 2.2% less than MobileNetv3.

**Discussion.** Combining convolutions and transformers results in robust and high-performing ViTs as compared to vanilla ViTs. However, an open question here is: *how to combine the strengths of convolutions and transformers to build light-weight networks for mobile vision tasks?* This paper focuses on designing light-weight ViT models that outperform state-of-the-art models with simple training recipes. Towards this end, we introduce MobileViT that combines the strengths of CNNs and ViTs to build a light-weight, general-purpose, and mobile-friendly network. MobileViT brings several novel observations. (i) **Better performance:** For a given parameter budget, MobileViT mod-

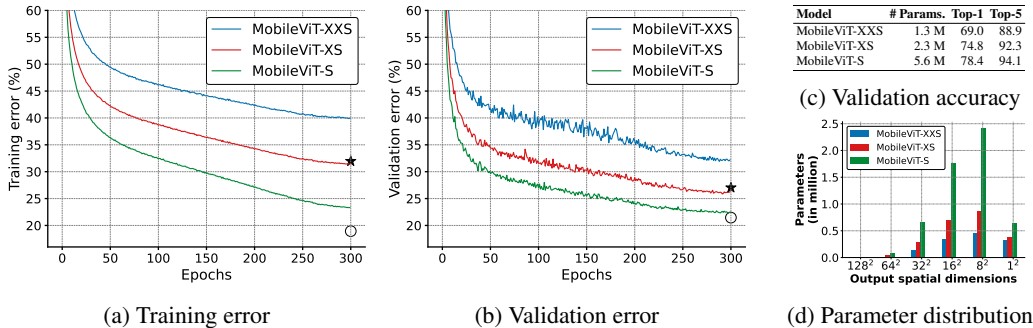

|           | (a) Training error | (b) Validation error | (d) Parameter distribution |
|-----------|--------------------|----------------------|----------------------------|

Figure 3: **MobileViT shows similar generalization capabilities as CNNs**. Final training and validation errors of MobileNetv2 and ResNet-50 are marked with ⋆ and ○, respectively (§B).

els achieve better performance as compared to existing light-weight CNNs across different mobile vision tasks (§4.1 and §4.2). (ii) **Generalization capability:** Generalization capability refers to the gap between training and evaluation metrics. For two models with similar training metrics, the model with better evaluation metrics is more generalizable because it can predict better on an unseen dataset. Unlike previous ViT variants (with and without convolutions) which show poor generalization capability even with extensive data augmentation as compared to CNNs (Dai et al., 2021), MobileViT shows better generalization capability (Figure 3). (iii) **Robust:** A good model should be robust to hyper-parameters (e.g., data augmentation and L2 regularization) because tuning these hyper-parameters is time- and resource-consuming. Unlike most ViT-based models, Mobile-ViT models train with basic augmentation and are less sensitive to L2 regularization (§C).

## 3 MobileViT: A Light-weight Transformer

A standard ViT model, shown in Figure 1a, reshapes the input $\mathbf{X} \in \mathbb{R}^{H \times W \times C}$ into a sequence of flattened patches $\mathbf{X}_f \in \mathbb{R}^{N \times PC}$, projects it into a fixed $d$-dimensional space $\mathbf{X}_p \in \mathbb{R}^{N \times d}$, and then learn inter-patch representations using a stack of $L$ transformer blocks. The computational cost of self-attention in vision transformers is $O(N^2 d)$. Here, $C$, $H$, and $W$ represent the channels, height, and width of the tensor respectively, and $P = wh$ is number of pixels in the patch with height $h$ and width $w$, and $N$ is the number of patches. Because these models ignore the spatial inductive bias that is inherent in CNNs, they require more parameters to learn visual representations. For instance, DPT (Dosovitskiy et al., 2021), a ViT-based network, learns $6\times$ more parameters as compared to DeepLabv3 (Chen et al., 2017), a CNN-based network, to deliver similar segmentation performance (DPT vs. DeepLabv3: 345 M vs. 59 M). Also, in comparison to CNNs, these models exhibit substandard optimizability. These models are sensitive to L2 regularization and require extensive data augmentation to prevent overfitting (Touvron et al., 2021a; Xiao et al., 2021).

This paper introduces a light-weight ViT model, MobileViT. The core idea is to learn global representations with transformers as convolutions. This allows us to implicitly incorporate convolution-like properties (e.g., spatial bias) in the network, learn representations with simple training recipes (e.g., basic augmentation), and easily integrate MobileViT with downstream architectures (e.g., DeepLabv3 for segmentation).

### 3.1 MobileViT Architecture

**MobileViT block.** The MobileViT block, shown in Figure 1b, aims to model the local and global information in an input tensor with fewer parameters. Formally, for a given input tensor $\mathbf{X} \in \mathbb{R}^{H \times W \times C}$, MobileViT applies a $n \times n$ standard convolutional layer followed by a point-wise (or $1 \times 1$) convolutional layer to produce $\mathbf{X}_L \in \mathbb{R}^{H \times W \times d}$. The $n \times n$ convolutional layer encodes local spatial information while the point-wise convolution projects the tensor to a high-dimensional space (or $d$-dimensional, where $d > C$) by learning linear combinations of the input channels.

With MobileViT, we want to model long-range non-local dependencies while having an effective receptive field of $H \times W$. One of the widely studied methods to model long-range dependencies

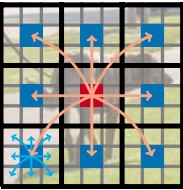

Figure 4: **Every pixel sees every other pixel in the MobileViT block.** In this example, the **red** pixel attends to **blue** pixels (pixels at the corresponding location in other patches) using transformers. Because **blue** pixels have already encoded information about the neighboring pixels using convolutions, this allows the **red** pixel to encode information from all pixels in an image. Here, each cell in **black** and **gray** grids represents a patch and a pixel, respectively.

is dilated convolutions. However, such approaches require careful selection of dilation rates. Otherwise, weights are applied to padded zeros instead of the valid spatial region (Yu & Koltun, 2016; Chen et al., 2017; Mehta et al., 2018). Another promising solution is self-attention (Wang et al., 2018; Ramachandran et al., 2019; Bello et al., 2019; Dosovitskiy et al., 2021). Among self-attention methods, vision transformers (ViTs) with multi-head self-attention are shown to be effective for visual recognition tasks. However, ViTs are heavy-weight and exhibit sub-standard optimizability. This is because ViTs lack spatial inductive bias (Xiao et al., 2021; Graham et al., 2021).

To enable MobileViT to learn global representations with spatial inductive bias, we unfold $\mathbf{X}_L$ into $N$ non-overlapping flattened patches $\mathbf{X}_U \in \mathbb{R}^{P \times N \times d}$. Here, $P = wh$, $N = \frac{HW}{P}$ is the number of patches, and $h \leq n$ and $w \leq n$ are height and width of a patch respectively. For each $p \in \{1, \cdots, P\}$, inter-patch relationships are encoded by applying transformers to obtain $\mathbf{X}_G \in \mathbb{R}^{P \times N \times d}$ as:

$$\mathbf{X}_G(p) = \text{Transformer}(\mathbf{X}_U(p)), 1 \leq p \leq P \tag{1}$$

Unlike ViTs that lose the spatial order of pixels, MobileViT neither loses the patch order nor the spatial order of pixels within each patch (Figure 1b). Therefore, we can fold $\mathbf{X}_G \in \mathbb{R}^{P \times N \times d}$ to obtain $\mathbf{X}_F \in \mathbb{R}^{H \times W \times d}$. $\mathbf{X}_F$ is then projected to low $C$-dimensional space using a point-wise convolution and combined with $\mathbf{X}$ via concatenation operation. Another $n \times n$ convolutional layer is then used to fuse these concatenated features. Note that because $\mathbf{X}_U(p)$ encodes local information from $n \times n$ region using convolutions and $\mathbf{X}_G(p)$ encodes global information across $P$ patches for the $p$-th location, each pixel in $\mathbf{X}_G$ can encode information from all pixels in $\mathbf{X}$, as shown in Figure 4. Thus, the overall effective receptive field of MobileViT is $H \times W$.

**Relationship to convolutions.** Standard convolutions can be viewed as a stack of three sequential operations: (1) unfolding, (2) matrix multiplication (to learn local representations), and (3) folding. MobileViT block is similar to convolutions in the sense that it also leverages the same building blocks. MobileViT block replaces the local processing (matrix multiplication) in convolutions with deeper global processing (a stack of transformer layers). As a consequence, MobileViT has convolution-like properties (e.g., spatial bias). Hence, the MobileViT block can be viewed as *transformers as convolutions*. An advantage of our intentionally simple design is that low-level efficient implementations of convolutions and transformers can be used out-of-the-box; allowing us to use MobileViT on different devices without any extra effort.

**Light-weight.** MobileViT block uses standard convolutions and transformers to learn local and global representations respectively. Because previous works (e.g., Howard et al., 2017; Mehta et al., 2021a) have shown that networks designed using these layers are heavy-weight, a natural question arises: Why MobileViT is light-weight? We believe that the issues lie primarily in learning global representations with transformers. For a given patch, previous works (e.g., Touvron et al., 2021a; Graham et al., 2021) convert the spatial information into latent by learning a linear combination of pixels (Figure 1a). The global information is then encoded by learning inter-patch information using transformers. As a result, these models lose image-specific inductive bias, which is inherent in CNNs. Therefore, they require more capacity to learn visual representations. Hence, they are deep and wide. Unlike these models, MobileViT uses convolutions and transformers in a way that the resultant MobileViT block has convolution-like properties while simultaneously allowing for global processing. This modeling capability allows us to design shallow and narrow MobileViT models, which in turn are light-weight. Compared to the ViT-based model DeIT that uses $L$=12 and $d$=192,

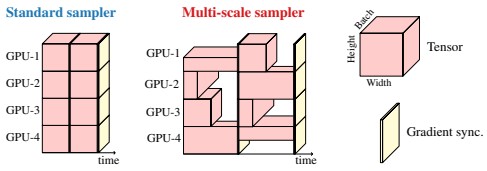

(a) Standard vs. multi-scale sampler illustration

| Sampler | # Updates | Epoch time |
|---|---|---|
| Standard | 375 k | 380 sec |
| Multi-scale (ours) | 232 k | 270 sec |

(b) Training efficiency. Here, **standard** sampler refers to PyTorch's Distributed-DataParallel sampler.

Figure 5: **Multi-scale vs. standard sampler.**

MobileViT model uses $L$= {2, 4, 3} and $d$={96, 120, 144} at spatial levels $32 \times 32$, $16 \times 16$, and $8 \times 8$, respectively. The resulting MobileViT network is faster ($1.85\times$), smaller ($2\times$), and better ($+1.8\%$) than DeIT network (Table 3; §4.3).

**Computational cost.** The computational cost of multi-headed self-attention in MobileViT and ViTs (Figure 1a) is $O(N^2Pd)$ and $O(N^2d)$, respectively. In theory, MobileViT is inefficient as compared to ViTs. However, in practice, MobileViT is more efficient than ViTs. MobileViT has $2\times$ fewer FLOPs and delivers $1.8\%$ better accuracy than DeIT on the ImageNet-1K dataset (Table 3; §4.3). We believe that this is because of similar reasons as for the light-weight design (discussed above).

**MobileViT architecture.** Our networks are inspired by the philosophy of light-weight CNNs. We train MobileViT models at three different network sizes (S: small, XS: extra small, and XXS: extra extra small) that are typically used for mobile vision tasks (Figure 3c). The initial layer in Mo-bileViT is a strided $3 \times 3$ standard convolution, followed by MobileNetv2 (or MV2) blocks and MobileViT blocks (Figure 1b and §A). We use Swish (Elfwing et al., 2018) as an activation function. Following CNN models, we use $n = 3$ in the MobileViT block. The spatial dimensions of feature maps are usually multiples of 2 and $h, w \le n$. Therefore, we set $h = w = 2$ at all spatial levels (see §C for more results). The MV2 blocks in MobileViT network are mainly responsible for down-sampling. Therefore, these blocks are shallow and narrow in MobileViT network. Spatial-level-wise parameter distribution of MobileViT in Figure 3d further shows that the contribution of MV2 blocks towards total network parameters is very small across different network configurations.

### 3.2 MULTI-SCALE SAMPLER FOR TRAINING EFFICIENCY

A standard approach in ViT-based models to learn multi-scale representations is *fine-tuning*. For instance, Touvron et al. (2021a) fine-tunes the DeIT model trained at a spatial resolution of $224 \times 224$ on varying sizes independently. Such an approach for learning multi-scale representations is preferable for ViTs because positional embeddings need to be interpolated based on the input size, and the network's performance is subjective to interpolation methods. Similar to CNNs, MobileViT does not require any positional embeddings and it may benefit from multi-scale inputs during training.

Previous CNN-based works (e.g., Redmon & Farhadi, 2017; Mehta et al., 2021b) have shown that multi-scale training is effective. However, most of these works sample a new spatial resolution after a fixed number of iterations. For example, YOLOv2 (Redmon & Farhadi, 2017) samples a new spatial resolution from a pre-defined set at every 10-th iteration and uses the same resolution across different GPUs during training. This leads to GPU under-utilization and slower training because the same batch size (determined using the maximum spatial resolution in the pre-defined set) is used across all resolutions. To facilitate MobileViT learn multi-scale representations without fine-tuning and to further improve training efficiency (i.e., fewer optimization updates), we extend the multi-scale training method to variably-sized batch sizes. Given a sorted set of spatial resolutions $\mathcal{S} = \{(H_1, W_1), \cdots, (H_n, W_n)\}$ and a batch size $b$ for a maximum spatial resolution of $(H_n, W_n)$, we randomly sample a spatial resolution $(H_t, W_t) \in \mathcal{S}$ at $t$-th training iteration on each GPU and compute the batch size for $t$-th iteration as: $b_t = \frac{H_n W_n b}{H_t W_t}$. As a result, larger batch sizes are used for smaller spatial resolutions. This reduces optimizer updates per epoch and helps in faster training.

Figure 5 compares standard and multi-scale samplers. Here, we refer to DistributedDataParallel in PyTorch as the standard sampler. Overall, the multi-scale sampler (i) reduces the training time as it requires fewer optimizer updates with variably-sized batches (Figure 5b), (ii) improves performance by about 0.5% (Figure 10; §B), and (iii) forces the network to learn better multi-scale representations (§B), i.e., the same network when evaluated at different spatial resolutions yields better performance

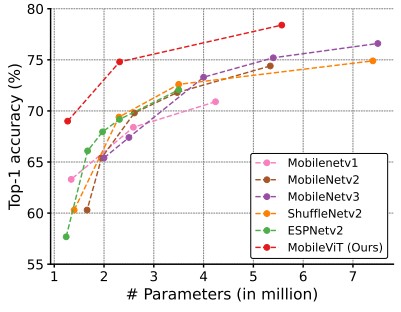

| Model | # Params. ⇓ | Top-1 ⇑ |
|---|---|---|
| MobileNetv1 | 2.6 M | 68.4 |
| MobileNetv2 | 2.6 M | 69.8 |
| MobileNetv3 | 2.5 M | 67.4 |
| ShuffleNetv2 | 2.3 M | 69.4 |
| ESPNetv2 | 2.3 M | 69.2 |
| MobileViT-XS (Ours) | 2.3 M | **74.8** |

(b) Comparison with light-weight CNNs (similar parameters)

| Model | # Params. ⇓ | Top-1 ⇑ |
|---|---|---|
| DenseNet-169 | 14 M | 76.2 |
| EfficientNet-B0 | 5.3 M | 76.3 |
| ResNet-101 | 44.5 M | 77.4 |
| ResNet-101-SE | 49.3 M | 77.6 |
| MobileViT-S (Ours) | 5.6 M | **78.4** |

(a) Comparison with light-weight CNNs     (c) Comparison with heavy-weight CNNs

Figure 6: **MobileViT vs. CNNs** on ImageNet-1k validation set. All models use basic augmentation.

as compared to the one trained with the standard sampler. In §B, we also show that the multi-scale sampler is generic and improves the performance of CNNs (e.g., MobileNetv2).

## 4 EXPERIMENTAL RESULTS

In this section, we first evaluate MobileViTs performance on the ImageNet-1k dataset and show that MobileViT delivers better performance than state-of-the-art networks (§4.1). In §4.2 and §4.3, we show MobileViTs are general-purpose and mobile-friendly, respectively.

### 4.1 IMAGE CLASSIFICATION ON THE IMAGENET-1K DATASET

**Implementation details.** We train MobileViT models from *scratch* on the ImageNet-1k classification dataset (Russakovsky et al., 2015). The dataset provides 1.28 million and 50 thousand images for training and validation, respectively. The MobileViT networks are trained using PyTorch for 300 epochs on 8 NVIDIA GPUs with an effective batch size of 1,024 images using AdamW optimizer (Loshchilov & Hutter, 2019), label smoothing cross-entropy loss (smoothing=0.1), and multi-scale sampler ($\mathcal{S} = \{(160, 160), (192, 192), (256, 256), (288, 288), (320, 320)\}$). The learning rate is increased from 0.0002 to 0.002 for the first 3k iterations and then annealed to 0.0002 using a cosine schedule (Loshchilov & Hutter, 2017). We use L2 weight decay of 0.01. We use basic data augmentation (i.e., random resized cropping and horizontal flipping) and evaluate the performance using a single crop top-1 accuracy. For inference, an exponential moving average of model weights is used.

**Comparison with CNNs.** Figure 6a shows that MobileViT outperforms *light-weight* CNNs across different network sizes (MobileNetv1 (Howard et al., 2017), MobileNetv2 (Sandler et al., 2018), ShuffleNetv2 (Ma et al., 2018), ESPNetv2 (Mehta et al., 2019), and MobileNetv3 (Howard et al., 2019)). For instance, for a model size of about 2.5 million parameters (Figure 6b), MobileViT outperforms MobileNetv2 by 5%, ShuffleNetv2 by 5.4%, and MobileNetv3 by 7.4% on the ImageNet-1k validation set. Figure 6c further shows that MobileViT delivers better performance than *heavy-weight* CNNs (ResNet (He et al., 2016), DenseNet (Huang et al., 2017), ResNet-SE (Hu et al., 2018), and EfficientNet (Tan & Le, 2019a)). For instance, MobileViT is 2.1% more accurate than EfficentNet for a similar number of parameters.

**Comparison with ViTs.** Figure 7 compares MobileViT with ViT variants that are trained from *scratch* on the ImageNet-1k dataset without distillation (DeIT (Touvron et al., 2021a), T2T (Yuan et al., 2021b), PVT (Wang et al., 2021), CAIT (Touvron et al., 2021b), DeepViT (Zhou et al., 2021), CeiT (Yuan et al., 2021a), CrossViT (Chen et al., 2021a), LocalViT (Li et al., 2021), PiT (Heo et al., 2021), ConViT (d'Ascoli et al., 2021), ViL (Zhang et al., 2021), BoTNet (Srinivas et al., 2021), and Mobile-former (Chen et al., 2021b)). Unlike ViT variants that benefit significantly from **advanced** augmentation (e.g., PiT w/ **basic** vs. **advanced**: 72.4 (R4) vs. 78.1 (R17); Figure 7b), MobileViT achieves better performance with fewer parameters and **basic** augmentation. For instance, MobileViT is $2.5\times$ smaller and $2.6\%$ better than DeIT (R3 vs. R8 in Figure 7b).

Overall, these results show that, similar to CNNs, MobileViTs are easy and robust to optimize. Therefore, they can be easily applied to new tasks and datasets.

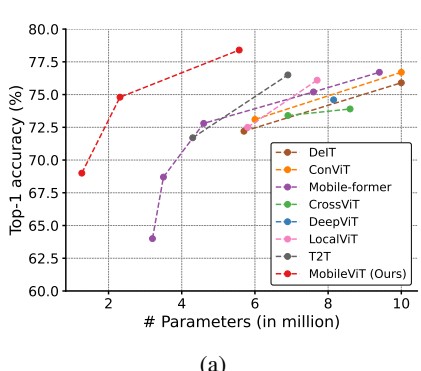

| Row # | Model | Augmentation | # Params. ⇓ | Top-1 ⇑ |
|---|---|---|---|---|
| R1 | DeIT | **Basic** | 5.7 M | 68.7 |
| R2 | T2T | **Advanced** | 4.3 M | 71.7 |
| R3 | DeIT | **Advanced** | 5.7 M | 72.2 |
| R4 | PiT | **Basic** | 10.6 M | 72.4 |
| R5 | Mobile-former | **Advanced** | 4.6 M | 72.8 |
| R6 | PiT | **Advanced** | 4.9 M | 73.0 |
| R7 | CrossViT | **Advanced** | 6.9 M | 73.4 |
| R8 | MobileViT-XS (Ours) | **Basic** | 2.3 M | **74.8** |
| R9 | CeiT | **Advanced** | 6.4 M | 76.4 |
| R10 | DeIT | **Advanced** | 10 M | 75.9 |
| R11 | T2T | **Advanced** | 6.9 M | 76.5 |
| R12 | ViL | **Advanced** | 6.7 M | 76.7 |
| R13 | LocalVit | **Advanced** | 7.7 M | 76.1 |
| R14 | Mobile-former | **Advanced** | 9.4 M | 76.7 |
| R15 | PVT | **Advanced** | 13.2 M | 75.1 |
| R16 | ConVit | **Advanced** | 10 M | 76.7 |
| R17 | PiT | **Advanced** | 10.6 M | 78.1 |
| R18 | BoTNet | **Basic** | 20.8 M | 77.0 |
| R19 | BoTNet | **Advanced** | 20.8 M | 78.3 |
| R20 | MobileViT-S (Ours) | **Basic** | 5.6 M | **78.4** |

(a)  (b)

Figure 7: **MobileViT vs. ViTs** on ImageNet-1k validation set. Here, **basic** means ResNet-style augmentation while **advanced** means a combination of augmentation methods with basic (e.g., MixUp (Zhang et al., 2018), RandAugmentation (Cubuk et al., 2019), and CutMix (Zhong et al., 2020)).

## 4.2 MOBILEVIT AS A GENERAL-PURPOSE BACKBONE

To evaluate the general-purpose nature of MobileViT, we benchmark MobileViT on two widely studied mobile vision tasks: (1) object detection (§4.2.1) and (2) semantic segmentation (§4.2.2).

### 4.2.1 MOBILE OBJECT DETECTION

**Implementation details.** We integrate MobileViT with a single shot object detection backbone (SSD; Liu et al., 2016). Following light-weight CNNs (e.g., MobileNets), we replace standard convolutions in the SSD head with separable convolutions and call the resultant network as SSDLite. We finetune MobileViT, pre-trained on the ImageNet-1k dataset, at an input resolution of $320 \times 320$ using AdamW on the MS-COCO dataset (Lin et al., 2014) that contains 117k training and 5k validation images. We use smooth L1 and cross-entropy losses for object localization and classification, respectively. The performance is evaluated on the validation set using mAP@IoU of 0.50:0.05:0.95. For other hyper-parameters, see §D.

| Feature backbone | # Params. ⇓ | mAP ⇑ |
|---|---|---|
| MobileNetv3 | 4.9 M | 22.0 |
| MobileNetv2 | 4.3 M | 22.1 |
| MobileNetv1 | 5.1 M | 22.2 |
| MixNet | 4.5 M | 22.3 |
| MNASNet | 4.9 M | 23.0 |
| MobileViT-XS (Ours) | **2.7 M** | 24.8 |
| MobileViT-S (Ours) | 5.7 M | **27.7** |

(a) Comparison w/ light-weight CNNs

| Feature backbone | # Params. ⇓ | mAP ⇑ |
|---|---|---|
| VGG | 35.6 M | 25.1 |
| ResNet50 | 22.9 M | 25.2 |
| MobileViT-S (Ours) | **5.7 M** | **27.7** |

(b) Comparison w/ heavy-weight CNNs

Table 1: **Detection w/ SSDLite.**

**Results.** Table 1a shows that, for the same input resolution of $320 \times 320$, SSDLite with MobileViT outperforms SSDLite with other light-weight CNN models (MobileNetv1/v2/v3, MNASNet, and MixNet). For instance, SSDLite's performance improves by 1.8%, and its model size reduces by $1.8\times$ when MobileViT is used as a backbone instead of MNASNet. Further, SSDLite with MobileViT outperforms standard SSD-300 with heavy-weight backbones while learning significantly fewer parameters (Table 1b). Also, qualitative results in §F confirms MobileViT's ability to detect variety of objects.

### 4.2.2 MOBILE SEMANTIC SEGMENTATION

**Implementation details.** We integrate Mobile-ViT with DeepLabv3 (Chen et al., 2017). We finetune MobileViT using AdamW with cross-entropy loss on the PASCAL VOC 2012 dataset (Everingham et al., 2015). Following a standard training practice (e.g., Chen et al., 2017; Mehta et al., 2019), we also use extra annotations and data from Hariharan et al. (2011) and Lin et al. (2014), respectively. The performance is evaluated on the validation set using mean intersection over union (mIOU). For other hyper-parameters, see §D.

| Feature backbone | # Params. ⇓ | mIOU ⇑ |
|---|---|---|
| MobileNetv1 | 11.2 M | 75.3 |
| MobileNetv2 | 4.5 M | 75.7 |
| MobileViT-XXS (Ours) | 1.9 M | 73.6 |
| MobileViT-XS (Ours) | 2.9 M | **77.1** |
| ResNet-101 | 58.2 M | **80.5** |
| MobileViT-S (Ours) | 6.4 M | 79.1 |

Table 2: **Segmentation w/ DeepLabv3.**

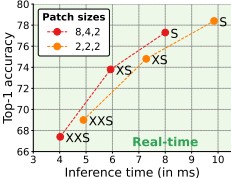 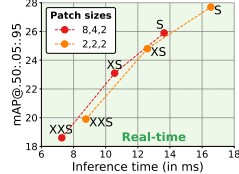 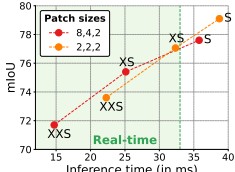

(a) Classification @ $256 \times 256$    (b) Detection @ $320 \times 320$    (c) Segmentation @ $512 \times 512$

Figure 8: **Inference time of MobileViT models on different tasks.** Here, dots in green color region represents that these models runs in real-time (inference time $< 33$ ms).

**Results.** Table 2 shows that DeepLabv3 with MobileViT is smaller and better. The performance of DeepLabv3 is improved by 1.4%, and its size is reduced by $1.6\times$ when MobileViT is used as a backbone instead of MobileNetv2. Also, MobileViT gives competitive performance to model with ResNet-101 while requiring $9\times$ fewer parameters; suggesting MobileViT is a powerful backbone. Also, results in §G shows that MobileViT learns generalizable representations of the objects and perform well on an *unseen* dataset.

## 4.3 PERFORMANCE ON MOBILE DEVICES

Light-weight and low latency networks are important for enabling mobile vision applications. To demonstrate the effectiveness of MobileViT for such applications, pre-trained full-precision Mobile-ViT models are converted to CoreML using publicly available CoreMLTools (2021). Their inference time is then measured (average over 100 iterations) on a mobile device, i.e., iPhone 12.

**Mobile-friendly.** Figure 8 shows the inference time of MobileViT networks with two patch size settings (Config-A: $2, 2, 2$ and Config-B: $8, 4, 2$) on three different tasks. Here $p_1, p_2, p_3$ in Config-X denotes the height $h$ (width $w = h$) of a patch at an output stride[2] of 8, 16, and 32, respectively. The models with smaller patch sizes (Config-A) are more accurate as compared to larger patches (Config-B). This is because, unlike Config-A models, Config-B models are not able to encode the information from all pixels (Figure 13 and §C). On the other hand, for a given parameter budget, Config-B models are faster than Config-A even though the theoretical complexity of self-attention in both configurations is the same, i.e., $\mathcal{O}(N^2 Pd)$. With larger patch sizes (e.g., $P=8^2=64$), we have fewer number of patches $N$ as compared to smaller patch sizes (e.g., $P=2^2=4$). As a result, the computation cost of self-attention is relatively less. Also, Config-B models offer a higher degree of parallelism as compared to Config-A because self-attention can be computed simultaneously for more pixels in a larger patch ($P=64$) as compared to a smaller patch ($P=4$). Hence, Config-B models are faster than Config-A. To further improve MobileViT's latency, linear self-attention (Wang et al., 2020) can be used. Regardless, all models in both configurations run in real-time (inference speed $\geq 30$ FPS) on a mobile device except for MobileViT-S models for the segmentation task. This is expected as these models process larger inputs ($512 \times 512$) as compared to classification ($256 \times 256$) and detection ($320 \times 320$) networks.

**Discussion.** We observe that MobileViT and other ViT-based networks (e.g., DeIT and PiT) are slower as compared to MobileNetv2 on mobile devices (Table 3). This observation contradicts previous works which show that ViTs are more scalable as compared to CNNs (Dosovitskiy et al., 2021). This difference is primarily because of two reasons. First, dedicated CUDA kernels exist for transformers on GPUs, which are used out-of-the-box in

| Model | # Params. ⇓ | FLOPs ⇓ | Time ⇓ | Top-1 ⇑ |
|---|---|---|---|---|
| MobileNetv2[†] | 3.5 M | **0.3 G** | **0.92 ms** | 73.3 |
| DeIT | 5.7 M | 1.3 G | 10.99 ms | 72.2 |
| PiT | 4.9 M | 0.7 G | 10.56 ms | 73.0 |
| MobileViT (Ours) | **2.3 M** | 0.7 G | 7.28 ms | **74.8** |

Table 3: **ViTs are slower than CNNs.** [†]Results with multi-scale sampler (§B).

ViTs to improve their scalability and efficiency on GPUs (e.g., Shoeybi et al., 2019; Lepikhin et al., 2021). Second, CNNs benefit from several device-level optimizations, including batch normalization fusion with convolutional layers (Jacob et al., 2018). These optimizations improve latency and memory access. However, such dedicated and optimized operations for transformers are currently not available for mobile devices. Hence, the resultant inference graph of MobileViT and ViT-based networks for mobile devices is sub-optimal. We believe that similar to CNNs, the inference speed of MobileViT and ViTs will further improve with dedicated device-level operations in the future.

---

[2]Output stride: Ratio of the spatial dimension of the input to the feature map.

## 5 ACKNOWLEDGEMENTS

We are grateful to Ali Farhadi, Peter Zatloukal, Oncel Tuzel, Ashish Shrivastava, Frank Sun, Max Horton, Anurag Ranjan, and anonymous reviewers for their helpful comments. We are also thankful to Apple's infrastructure and open-source teams for their help with training infrastructure and open-source release of the code and pre-trained models.

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

## A   MOBILEVIT ARCHITECTURE

MobileViT's are inspired by the philosophy of light-weight CNNs and the overall architecture of MobileViT at different parameter budgets is given in Table 4. The initial layer in MobileViT is a strided $3 \times 3$ standard convolution, followed by MobileNetv2 (or MV2) blocks and MobileViT blocks. We use Swish (Elfwing et al., 2018) as an activation function. Following CNN models, we use $n = 3$ in the MobileViT block. The spatial dimensions of feature maps are usually multiples of 2 and $h, w \leq n$. Therefore, we set $h = w = 2$ at all spatial levels. The MV2 blocks in MobileViT network are mainly responsible for down-sampling. Therefore, in these blocks, we use an expansion factor of four, except for MobileViT-XXS where we use an expansion factor of 2. The transformer layer in MobileViT takes a $d$-dimensional input, as shown in Figure 1b. We set the output dimension of the first feed-forward layer in a transformer layer as $2d$ instead of $4d$, a default value in the standard transformer block of Vaswani et al. (2017).

## B   MULTI-SCALE SAMPLER

**Multi-scale sampler reduces generalization gap.** Generalization capability refers to the gap between training and evaluation metrics. For two models with similar training metrics, the model with better evaluation metrics is more generalizable because it can predict better on an unseen dataset. Figure 9a and Figure 9b compares the training and validation error of the MobileViT-S model trained with standard and multi-scale samplers. The training error of MobileViT-S with multi-scale sampler is higher than standard sampler while validation error is lower. Also, the gap between training error and validation error of MobileViT-S with multi-scale sampler is close to zero. This suggests that a multi-scale sampler improves generalization capability. Also, when MobileViT-S trained independently with standard and multi-scale sampler is evaluated at different input resolutions (Figure 9c), we observe that MobileViT-S trained with multi-scale sampler is more robust as compared to the one trained with the standard sampler. We also observe that multi-scale sampler improves the performance of MobileViT models at different model sizes by about 0.5% (Figure 10). These observations in conjunction with impact on training efficiency (Figure 5b) suggests that a multi-scale sampler is effective. Pytorch implementation of multi-scale sampler is provided in Listing 1.

**Multi-scale sampler is generic.** We train a heavy-weight (ResNet-50) and a light-weight (MobileNetv2-1.0) CNN with the multi-scale sampler to demonstrate its generic nature. Results in Table 5 show that a multi-scale sampler improves the performance as well as training efficiency.

| Layer | Output size | Output stride | Repeat | Output channels | | |
|---|---|---|---|---|---|---|
| | | | | XXS | XS | S |
| Image | $256 \times 256$ | 1 | | | | |
| Conv-$3 \times 3, \downarrow 2$ | $128 \times 128$ | 2 | 1 | 16 | 16 | 16 |
| MV2 | | | 1 | 16 | 32 | 32 |
| MV2, $\downarrow 2$ | $64 \times 64$ | 4 | 1 | 24 | 48 | 64 |
| MV2 | | | 2 | 24 | 48 | 64 |
| MV2, $\downarrow 2$ | $32 \times 32$ | 8 | 1 | 48 | 64 | 96 |
| MobileViT block ($L = 2$) | | | 1 | 48 ($d = 64$) | 64 ($d = 96$) | 96 ($d = 144$) |
| MV2, $\downarrow 2$ | $16 \times 16$ | 16 | 1 | 64 | 80 | 128 |
| MobileViT block ($L = 4$) | | | 1 | 64 ($d = 80$) | 80 ($d = 120$) | 128 ($d = 192$) |
| MV2, $\downarrow 2$ | $8 \times 8$ | 32 | 1 | 80 | 96 | 160 |
| MobileViT block ($L = 3$) | | | 1 | 80 ($d = 96$) | 96 ($d = 144$) | 160 ($d = 240$) |
| Conv-$1 \times 1$ | | | 1 | 320 | 384 | 640 |
| Global pool | $1 \times 1$ | 256 | 1 | | | |
| Linear | | | | 1000 | 1000 | 1000 |
| **Network Parameters** | | | | 1.3 M | 2.3 M | 5.6 M |

Table 4: **MobileViT architecture.** Here, $d$ represents dimensionality of the input to the transformer layer in MobileViT block (Figure 1b). By default, in MobileViT block, we set kernel size $n$ as three and spatial dimensions of patch (height $h$ and width $w$) in MobileViT block as two.

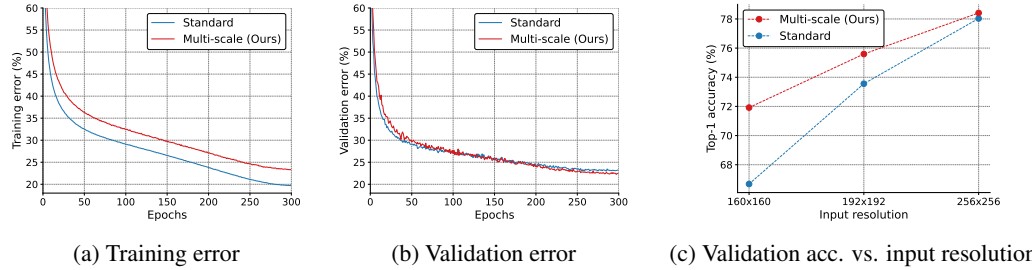

(a) Training error       (b) Validation error       (c) Validation acc. vs. input resolution

Figure 9: **MobileViT-S learns better representations with multi-scale sampler on ImageNet-1k.**

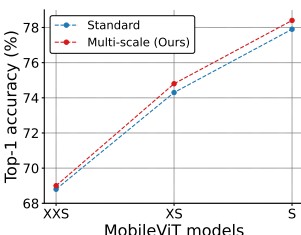

Figure 10: **MobileViT's performance on ImageNet-1k with standard and multi-scale sampler.**

| Model | # Params | # Epochs | # Updates ⇓ | Top-1 accuracy ⇑ | | Training time ⇓ | |
|---|---|---|---|---|---|---|---|
| ResNet-50 w/ standard sampler (PyTorch) | 25 M | – | – | 76.2 | (0.0%) | – | – |
| ResNet-50 w/ standard sampler (our repro.)[†] | 25 M | 150 | 187 k | 77.1 | (+0.9%) | 54 k sec. | (1.35×) |
| ResNet-50 w/ multi-scale sampler (Ours)[†] | 25 M | 150 | 116 k | **78.6** | (+2.4%) | 40 k sec. | (1×) |
| MobileNetv2-1.0 w/ standard sampler (PyTorch) | 3.5 M | – | – | 71.9 | (0.0%) | – | (–) |
| MobileNetv2-1.0 w/ standard sampler (our repro.)[†] | 3.5 M | 300 | 375 k | 72.1 | (+0.2%) | 78 k sec. | (1.16×) |
| MobileNetv2-1.0 w/ multi-scale sampler (Ours)[†] | 3.5 M | 300 | 232 k | **73.3** | (+1.4%) | 67 k sec. | (1×) |

Table 5: **Multi-scale sampler is generic.** All models are trained with basic augmentation on the ImageNet-1k. [†]Results are with exponential moving average.

For instance, a multi-scale sampler improves the performance of MobileNetv2-1.0 by about 1.4% while decreasing the training time by 14%.

## C ABLATIONS

**Impact of weight decay.** A good model should be insensitive or less sensitive to L2 regularization (or weight decay) because tuning it for each task and dataset is time- and resource-consuming. Unlike CNNs, ViT models are sensitive to weight decay (Dosovitskiy et al., 2021; Touvron et al., 2021a; Xiao et al., 2021). To study if MobileViT models are sensitive to weight decay or not, we train the MobileViT-S model by varying the value of weight decay from 0.1 to 0.0001. Results are shown in Figure 11. With an exception to the MobileViT model trained with a weight decay of 0.1, all other models converged to a similar solution. This shows that MobileViT models are robust to weight decay. In our experiments, we use the value of weight decay as 0.01. Note that 0.0001 is the widely used value of weight decay in most CNN-based models, such as ResNet and DenseNet. Even at this value of weight decay, MobileViT outperforms CNNs on the ImageNet-1k dataset (e.g., DenseNet vs. MobileViT: 76.2 with 14 M parameters vs. 77.4 with 5.7 M parameters).

**Impact of skip-connection.** Figure 12 studies the impact of skip-connection in the MobileViT block (**red** arrow in Figure 1b). With this connection, the performance of MobileViT-S improves by 0.5% on the ImageNet dataset. Note that even without this skip-connection, MobileViT-S delivers similar or better performance than state-of-the-art CNN- (Figure 6) and ViT-based (Figure 7b) models, that too with basic data augmentation.

**Impact of patch sizes.** MobileViT combines convolutions and transformers to learn local and global representations effectively. Because convolutions are applied on $n \times n$ regions and self-attention

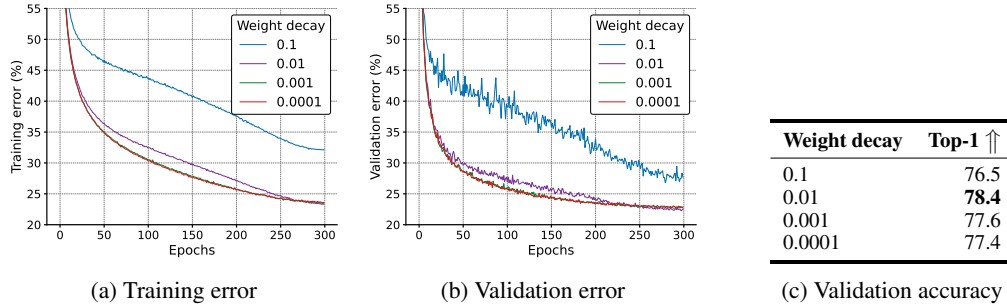

| | (a) Training error | (b) Validation error | (c) Validation accuracy |

| Weight decay | Top-1 ⇑ |
| --- | --- |
| 0.1 | 76.5 |
| 0.01 | **78.4** |
| 0.001 | 77.6 |
| 0.0001 | 77.4 |

Figure 11: **Impact of weight decay**. Here, results are shown for MobileViT-S model (5.7 M parameters) on the ImageNet-1k dataset. Results in **(c)** are with exponential moving average.

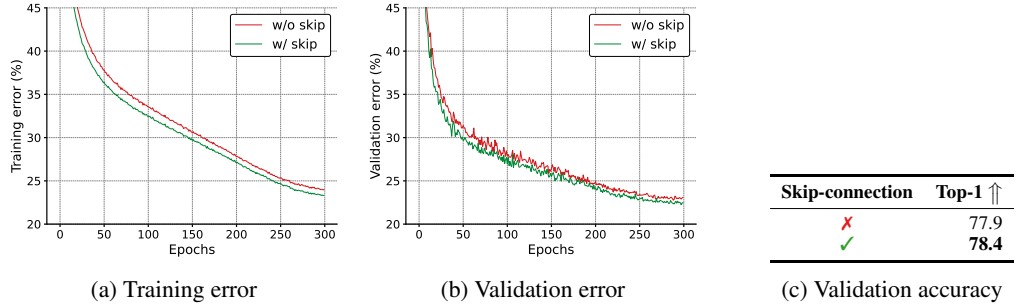

| | (a) Training error | (b) Validation error | (c) Validation accuracy |

| Skip-connection | Top-1 ⇑ |
| --- | --- |
| ✗ | 77.9 |
| ✓ | **78.4** |

Figure 12: **Impact of skip connection**. Here, results are shown for MobileViT-S model (5.7 M parameters) on the ImageNet-1k dataset. Results in **(c)** are with exponential moving average.

| Patch sizes | # Params. | Time ⇓ | Top-1 ⇑ |
| --- | --- | --- | --- |
| 2,2,2 | 5.7 M | 9.85 ms | 78.4 |
| 3,3,3$^\dagger$ | 5.7 M | 14.69 ms | **78.5** |
| 4,4,4 | 5.7 M | 8.23 ms | 77.6 |
| 8,4,2 | 5.7 M | 8.20 ms | 77.3 |

Table 6: **Impact of patch sizes**. Here, the patch sizes are for spatial levels at $32 \times 32$, $16 \times 16$, and $8 \times 8$, respectively. Also, results are shown for MobileViT-S model on the ImageNet-1k dataset. Results are with exponential moving average. $^\dagger$ Spatial dimensions of feature map are not multiple of patch dimensions. Therefore, we use bilinear interpolation in folding and unfolding operations to resize the feature map.

is computed over patches with spatial dimensions of $h$ and $w$, it is essential to establish a good relationship between $n$, $h$, and $w$. Following previous works on CNN designs, we set $n = 3$ and then vary $h$ and $w$. Specifically, we study four configurations: (i) $h = w = 2$ at all spatial levels (Figure 13a). In this case, $h, w < n$ and would allow each pixel to encode information from every other pixel using MobileViT. (ii) $h = w = 3$ at all spatial levels (Figure 13b). In this case, $h = w = n$. Similar to (i), this configuration would also allow each pixel to encode information from every other pixel using MobileViT. (iii) $h = w = 4$ at all spatial levels (Figure 13c). In this case, $h, w > n$ and would not allow each pixel to aggregate information from other pixels in the tensor. (iv) $h = w = 8$, $h = w = 4$, and $h = w = 2$ at spatial level of $32 \times 32$, $16 \times 16$, and $8 \times 8$, respectively. Unlike (i), (ii), and (iii), the number of patches $N$ is the same across different spatial resolutions in (iv). Also, $h, w < n$ only for a spatial level of $8 \times 8$ where $h = w = 2$. Note that all these models have the same number of network parameters and the same computational cost of self-attention, i.e., $\mathcal{O}(N^2 Pd)$. Here, $N$ is the number of patches, $P = hw$ is the number of pixels in a patch with height $h$ and width $w$, and $d$ is the model dimension.

Results are shown in Table 6. We can see that when $h, w \leq n$, MobileViT can aggregate information more effectively, which helps improve performance. In our experiments, we used $h = w = 2$ instead of $h = w = 3$ because spatial dimensions of feature maps are multiples of 2, and using

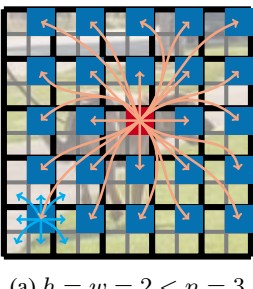 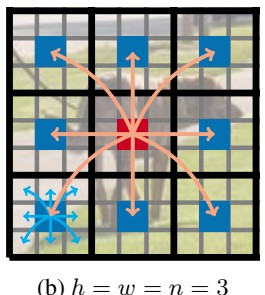 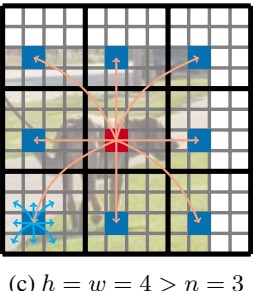

(a) $h = w = 2 < n = 3$    (b) $h = w = n = 3$    (c) $h = w = 4 > n = 3$

Figure 13: **Relationship between kernel size ($n \times n$) for convolutions and patch size ($h \times w$) for folding and unfolding in MobileViT.** In **a** and **b**, the **red** pixel is able to aggregate information from all pixels using local (**cyan** colored arrows) and global (**orange** colored arrows) information while in **(c)**, every pixel is not able to aggregate local information using convolutions with kernel size of $3 \times 3$ from $4 \times 4$ patch region. Here, each cell in **black** and **gray** grids represents a patch and pixel, respectively.

| LS | EMA | Top-1 ⇑ |
|:---:|:---:|:---:|
| ✗ | ✗ | 78.0 |
| ✓ | ✗ | 78.3 |
| ✓ | ✓ | **78.4** |

Table 7: **Effect of label smoothing (LS) and exponential moving average (EMA) on the performance of MobileViT-S on the ImageNet-1k dataset.** First row results are with cross-entropy.

$h = w = 3$ requires additional operations. For folding and unfolding, we need to either pad or resize. In the case of padding, we need to mask the padded pixels in self-attention in transformers. These additional operations result in latency, as shown in Table 6. To avoid these extra operations, we choose $h = w = 2$ in our experiments, which also provides a good trade-off between latency and accuracy.

**Impact of exponential moving average and label smoothing.** Exponential moving average (EMA) and label smoothing (LS) are two standard training methods that are used to improve CNN- and Transformer-based models performance (Sandler et al., 2018; Howard et al., 2019; Tan et al., 2019; Touvron et al., 2021a; Dai et al., 2021; Xiao et al., 2021). Table 7 shows that LS marginally improves the performance of MobileViT-S while EMA has little or no effect on model's performance on the ImageNet-1k dataset. Because previous works have shown these methods to be effective in reducing stochastic noise and prevent network from becoming over-confident, we use these methods to train MobileViT models.

# D    TRAINING DETAILS FOR SSDLITE AND DEEPLABV3

All SSDLite-MobileViT and DeepLabv3-MobileViT networks are trained for 200 and 50 epochs with a standard sampler on 4 NVIDIA GPUs and with an effective batch size of 128 images, respectively. The learning rate is increased from 0.00009 to 0.0009 in the first 500 iterations and then annealed to 0.00009 using a cosine learning rate scheduler. We use L2 weight decay of 0.01. We change the stride of MV2 block from two to one at an output stride of 32 in Table 4 to obtain DeepLabv3-MobileViT  models at an output stride of 16.

For these models, we do not use a multi-scale sampler. This is because these task-specific networks are resolution-dependent. For example, DeepLabv3 uses an atrous (or dilation) rate of 6, 12, and 18 at an output stride of 16 to learn multi-scale representations. If we use a lower resolution (say $256 \times 256$) than $512 \times 512$, then the atrous kernel weights will be applied to padded zeros; making multi-scale learning ineffective.

# E    EXTENDED DISCUSSION

**Memory footprint.**    A light-weight network running on mobile devices should be memory efficient. Similar to MobileNetv2, we measure the memory that needs to be materialized at each spatial level (Table 8). At lower spatial levels (i.e., an output stride of 8, 16, and 32) where MobileViT blocks are employed, required memory is lesser or comparable to light-weight CNNs. Therefore, similar to light-weight CNNs, MobileViT networks are also memory efficient.

**FLOPs.**    Floating point operations (FLOPs) is another metric that is widely used to measure the efficiency of a neural network. Table 9 compare FLOPs of MobileViT with different ViT-based networks on the ImageNet-1k dataset. For similar number of FLOPs, MobileViT is faster, smaller, and better. For instance, PiT and MobileViT has the same number of FLOPs, but MobileViT is $1.45\times$ faster, $2.1\times$ smaller, and $1.8\%$ better (R2 vs. R4 in Table 9). It is important to note that FLOPs for networks in R2-R4 are the same, but their latency and performance are different. This shows that FLOPs is not a sufficient metric for network efficiency as it does not account for inference-related factors such as memory access, degree of parallelism, and platform characteristics.

The ImageNet-1k pre-training helps in performance improvement in down-stream tasks such as object detection and semantic segmentation (Long et al., 2015; Chen et al., 2017; Redmon & Farhadi, 2017). Because such tasks are used in real-world applications and often uses higher image inputs as compared to the ImageNet-1k classification task, it is important to compare the FLOPs of a network on down-stream tasks. Towards this end, we compare the FLOPs of MobileViT with MobileNetv2 on three tasks, i.e., classification, detection, and segmentation. Results are shown in Table 10. We can observe that (1) the gap between MobileNetv2 and MobileViT FLOPs reduces as the input resolution increases. For instance, MobileNetv2 has $2\times$ fewer FLOPs as compared to MobileViTon the ImageNet-1k classification task, but on the semantic segmentation, they have similar FLOPs (Table 10a vs. Table 10c) and (2) MobileNetv2 models are significantly faster but less accurate than MobileViT models across different tasks. The low-latency of MobileNetv2 models is likely because of dedicated and optimized hardware-level operations on iPhone. We believe that (1) the inference speed of MobileViT will further improve with such dedicated operations and (2) our results will inspire future research in the area of hardware design and optimization.

**Inference time on different devices.**    Table 11 compares the inference time of different models on three different devices, i.e., iPhone12 CPU, iPhone12 neural engine, and NVIDIA V100 GPU. MobileNetv2 is the fastest network across all devices. On iPhone (both CPU and neural engine), MobileViT delivers better performance as compared to DeIT and PiT. However, on GPU, DeIT and PiT are faster than MobileViT. This is likely because MobileViT models (1) are shallow and narrow, (2) run at higher spatial resolution ($256 \times 256$ instead of $224 \times 224$), and (2) did not use

| OS | MobileNetv2-1.0 | MobileViT-XS |
|---|---|---|
| 2 | 400 | 784 |
| 4 | 200 | 294 |
| 8 | 100 | 98 |
| 16 | 62 | 31 |
| 32 | 32 | 37 |
| **Top-1** | 73.3 | 74.8 |

Table 8: **Comparison between MobileNetv2 and MobileViT in terms of maximum memory (in kb) that needs to be materialized at each spatial resolution in the network.** The top-1 accuracy is measured on the ImageNet-1k validation set. Here, OS (output stride) is the ratio of spatial dimensions of the input to the feature map.

| Model | # Params. ⇓ | FLOPs ⇓ | Time ⇓ | Top-1 ⇑ |
|---|---|---|---|---|
| (R1) DeIT | 5.7 M | 1.3 G | 10.99 ms | 72.2 |
| (R2) PiT | 4.9 M | **0.7 G** | 10.56 ms | 73.0 |
| (R3) MobileViT-XS (Ours; 8,4,2) | **2.3 M** | **0.7 G** | **5.93 ms** | 73.8 |
| (R4) MobileViT-XS (Ours; 2,2,2) | **2.3 M** | **0.7 G** | 7.28 ms | **74.8** |

Table 9: **Comparison of different ViT-based networks**. The performance of MobileViT-XS model is reported at two different patch-size settings. See §A for details.

| Model | # Params. ⇓ | FLOPs ⇓ | Time ⇓ | Top-1 ⇑ |
|---|---|---|---|---|
| MobileNetv2 | 3.5 M | **0.3 G** | **0.92 ms** | 73.3 |
| MobileViT-XS (Ours; 8,4,2) | **2.3 M** | 0.7 G | 5.93 ms | 73.8 |
| MobileViT-XS (Ours; 2,2,2) | **2.3 M** | 0.7 G | 7.28 ms | **74.8** |

(a) ImageNet-1k classification

| Backbone | # Params. ⇓ | FLOPs ⇓ | Time ⇓ | mAP ⇑ |
|---|---|---|---|---|
| MobileNetv2 | 4.3 M | 0.8 G | **2.3 ms** | 22.1 |
| MobileViT-XS (Ours; 8,4,2) | **2.7 M** | 1.6 G | 10.7 ms | 23.1 |
| MobileViT-XS (Ours;2,2,2) | **2.7 M** | 1.6 G | 12.6 ms | **24.8** |

(b) Object detection w/ SSDLite.

| Backbone | # Params. ⇓ | FLOPs ⇓ | Time ⇓ | mIOU ⇑ |
|---|---|---|---|---|
| MobileNetv2 | 4.3 M | 5.8 G | **6.5 ms** | 75.7 |
| MobileViT-XS (Ours) | **2.9 M** | **5.7 G** | 25.1 ms | 75.4 |
| MobileViT-XS (Ours) | **2.9 M** | **5.7 G** | 32.3 ms | **77.1** |

(c) Semantic segmentation w/ DeepLabv3.

Table 10: **MobileViT vs. MobileNetv2 on different tasks**. The FLOPs and inference time in (a), (b) and (c) are measured at $224 \times 224$, $320 \times 320$, and $512 \times 512$ respectively with an exception to MobileViT-XS model in (a) which uses $256 \times 256$ as an input resolution for measuring inference time on iPhone 12 neural engine. Here, the performance of MobileViT-XS models is reported at two different patch-size settings. See §A for details.

| Model | # Params ⇓ | FLOPs ⇓ | Top-1 ⇑ | Inference time ⇓ | | |
|---|---|---|---|---|---|---|
| | | | | iPhone12 - CPU | iPhone12 - Neural Engine | NVIDIA V100 GPU |
| MobileNetv2 | 3.5 M | **0.3 G** | 73.3 | **7.50 ms** | **0.92 ms** | **0.31 ms** |
| DeiT | 5.7 M | 1.3 G | 72.2 | 28.15 ms | 10.99 ms | 0.43 ms |
| PiT | 4.9 M | 0.7 G | 73.0 | 24.03 ms | 10.56 ms | 0.46 ms |
| MobileViT (Ours) | **2.3 M** | 0.7 G | **74.8** | 17.86 ms | 7.28 ms | 0.62 ms/0.47 ms[†] |

Table 11: **Inference time on different devices.** The run time of MobileViT is measured at $256 \times 256$ while for other networks, it is measured at $224 \times 224$. For GPU, inference time is measured for a batch of 32 images while for other devices, we use a batch size of one. Here, † represents that MobileViT model uses PyTorch's Unfold and Fold operations. Also, patch sizes for MobileViT model at an output stride of 8, 16, and 32 are set to two.

GPU- accelerated operations for folding and unfolding as they are not supported on mobile devices. However, when we replaced our *unoptimized* fold and unfold operations with PyTorch's Unfold and Fold operations, the latency of MobileViT model is improved from 0.62 ms to 0.47 ms.

Overall, our findings suggest that they are opportunities for optimizing ViT-based models, including MobileViT, for different accelerators. We believe that our work will inspire future research in building more efficient networks.

# F  QUALITATIVE RESULTS ON THE TASK OF OBJECT DETECTION

Figures 15, 14, and 16 shows that SSDLite with MobileViT-S can detect different objects under different settings, including changes in illumination and viewpoint, different backgrounds, and non-rigid deformations.

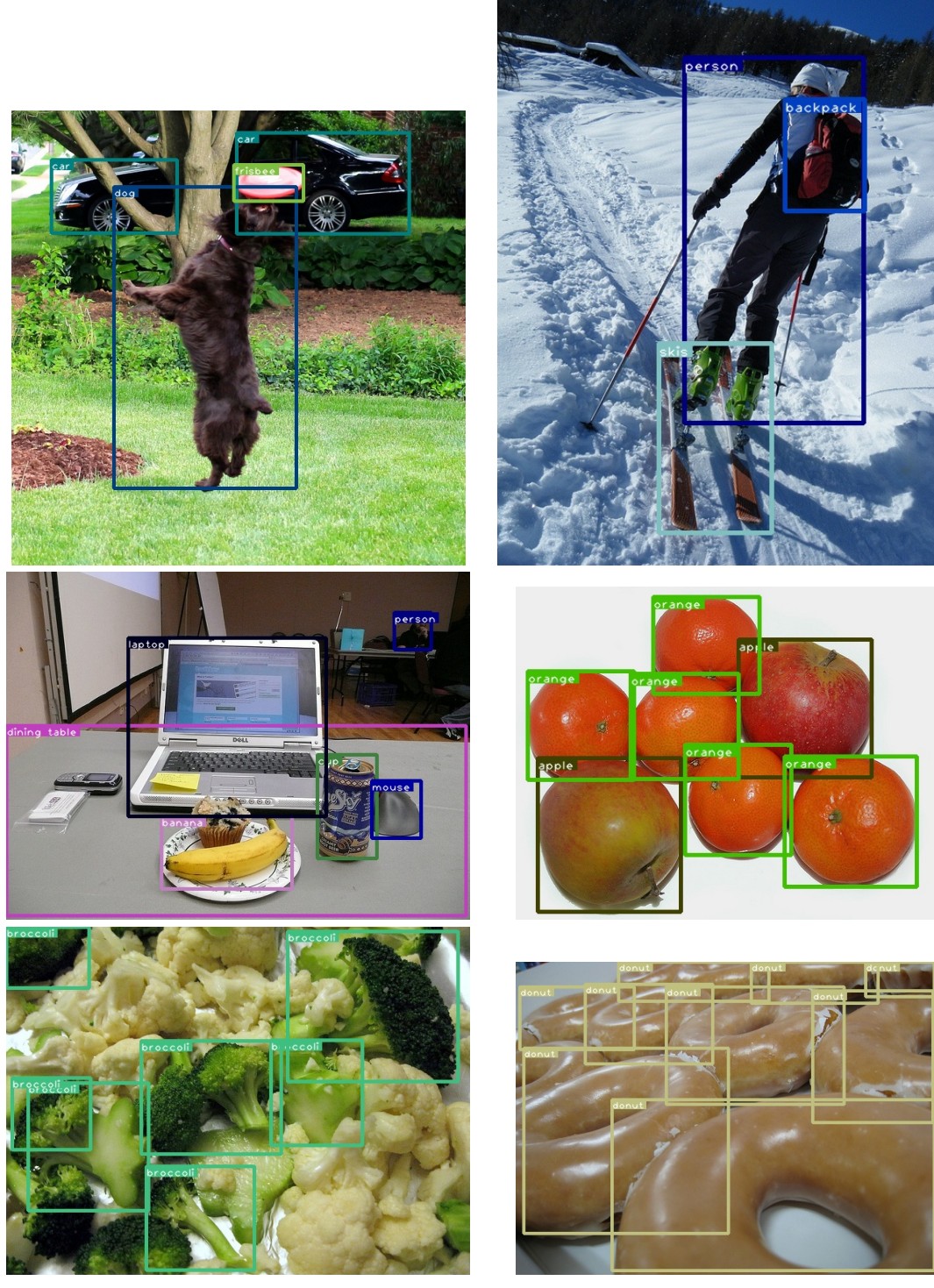

Figure 14: **Object detection results** of SSDLite-MobileViT-S on the MS-COCO validation set.

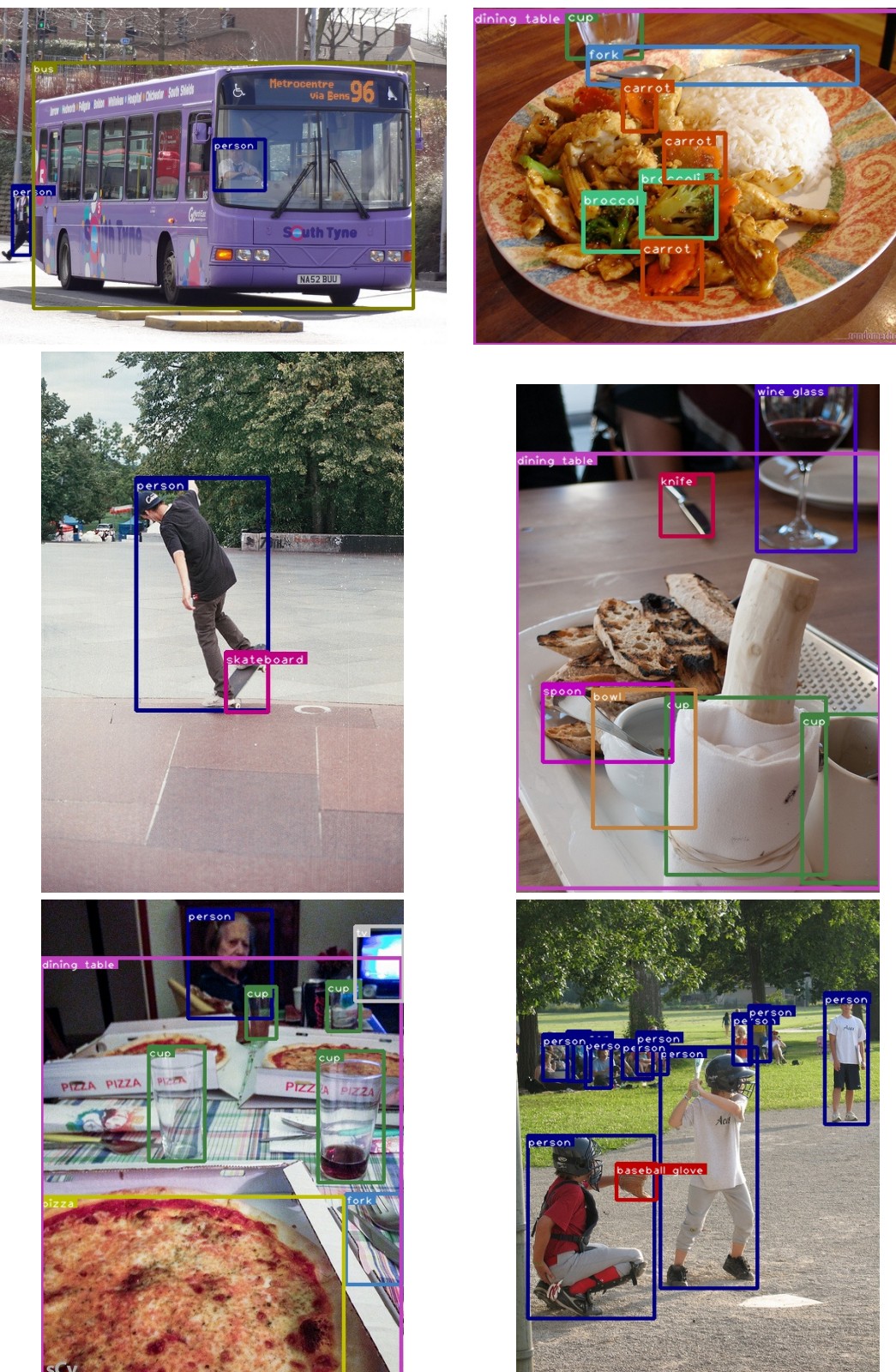

Figure 15: **Object detection results** of SSDLite-MobileViT-S on the MS-COCO validation set.

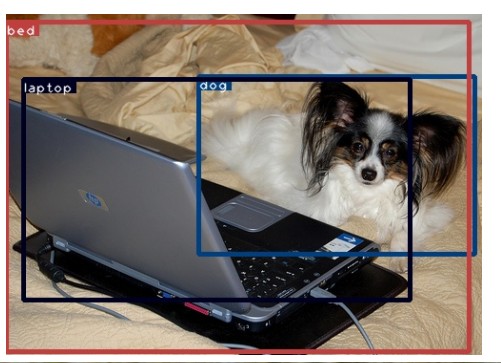
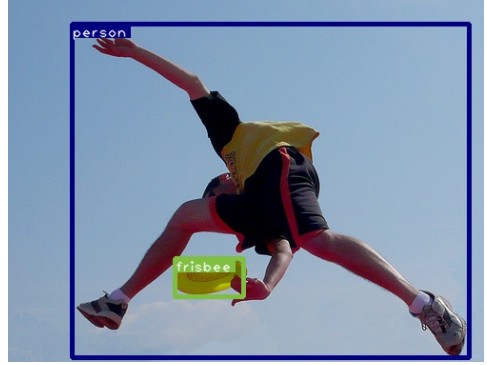
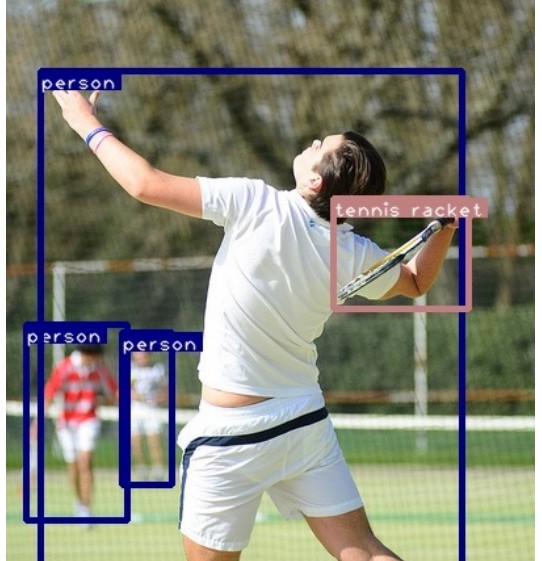
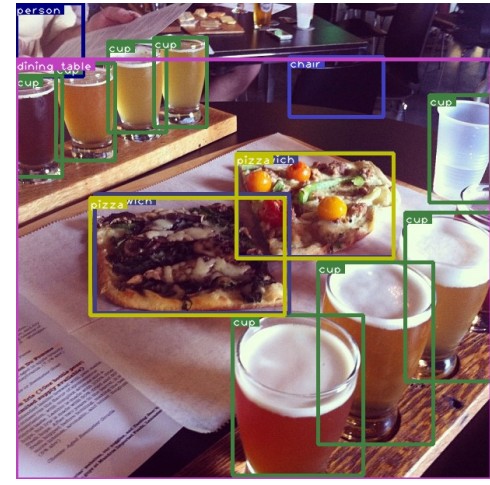
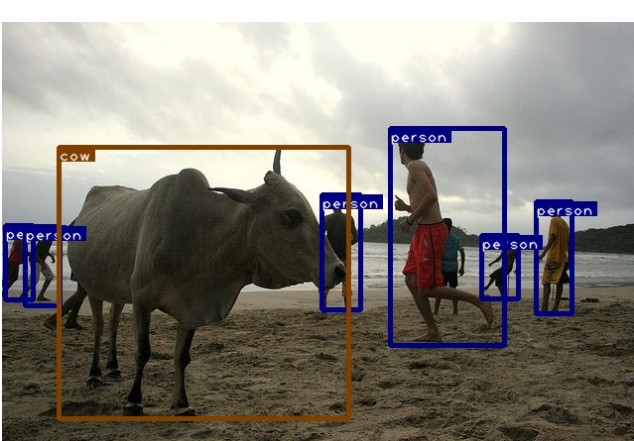
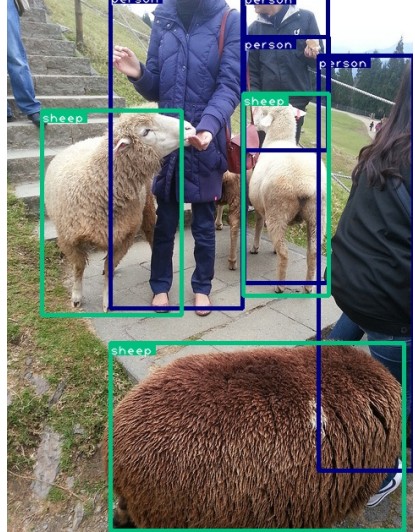

Figure 16: **Object detection results** of SSDLite-MobileViT-S on the MS-COCO validation set.

# G  SEMANTIC SEGMENTATION RESULTS ON AN UNSEEN DATASET

To demonstrate that MobileViT learns good generalizable representations of objects, we evaluate the DeepLabv3-MobileViT model in Section 4.2.2 on the MS-COCO validation set that contains 5k images. Following official torchvision segmentation models (PyTorch, 2021), object classes in the MS-COCO dataset are mapped to the object classes in the PASCAL VOC dataset and models are evaluated in terms of mIOU. Note that the MS-COCO validation set is an *unseen* test set for DeepLabv3-MobileViT models because these images are neither part of the training nor the validation set used for training DeepLabv3-MobileViT models.

Table 12 compares the performance of DeepLabv3-MobileViT models with MobileNetv3-Large that was trained with three different segmentation backbones (LR-ASPP (Howard et al., 2019), DeepLabv3, and FCN (Long et al., 2015)). For the same segmentation model, i.e., DeepLabv3, MobileViT is a more effective backbone than MobileNetv3. DeepLabv3-MobileViT-S model is 1.7× smaller and 5.1% more accurate than DeepLabv3-MobileNetv3-Large model. Furthermore, qualitative results in Figure 17 and Figure 18 further demonstrates that MobileViT learns good generalizable representations of the objects and perform well in the wild.

| Model | # Params $\Downarrow$ | mIOU $\Uparrow$ |
|---|---|---|
| LR-ASPP w/ MobileNetV3-Large | 3.2 M | 57.9 |
| FCN w/ MobileNetV3-Large | 5.1 M | 57.8 |
| DeepLabv3 w/ MobileNetV3-Large | 11.0 M | 60.3 |
| DeepLabv3 w/ MobileViT-XXS (Ours) | **1.9 M** | 46.7 |
| DeepLabv3 w/ MobileViT-XS (Ours) | 2.9 M | 57.4 |
| DeepLabv3 w/ MobileViT-S (Ours) | 6.4 M | **65.4** |

Table 12: **Semantic segmentation on the MS-COCO validation set.** MobileNetv3-Large results are from official torchvision segmentation models (PyTorch, 2021).

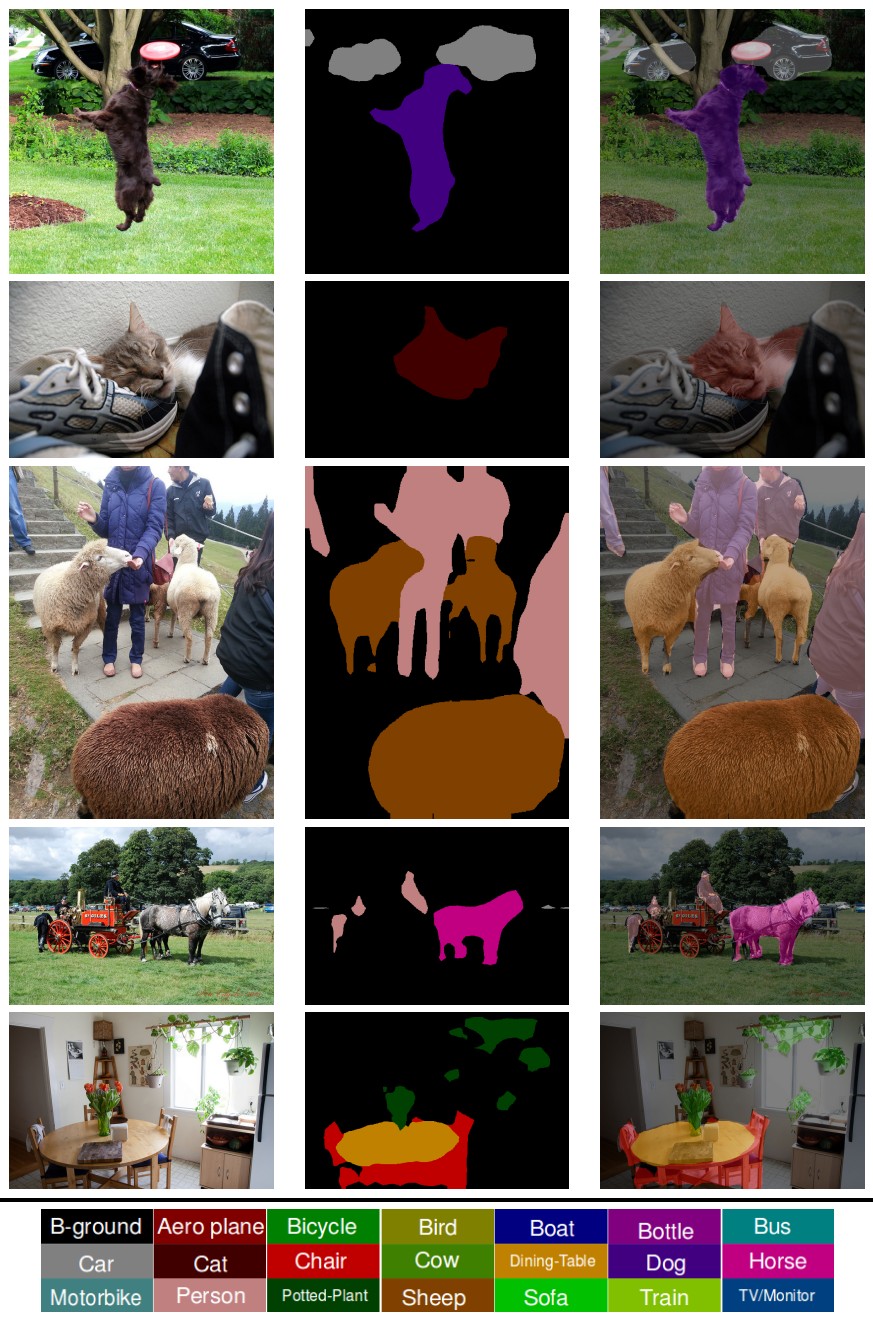

Figure 17: **Semantic segmentation results** of Deeplabv3-MobileViT-S model on the *unseen* MS-COCO validation set (**left:** input RGB image, **middle:** predicted segmentation mask, and **right:** Segmentation mask overlayed on RGB image). Color encoding for different objects in the PASCAL VOC dataset is shown in the last row.

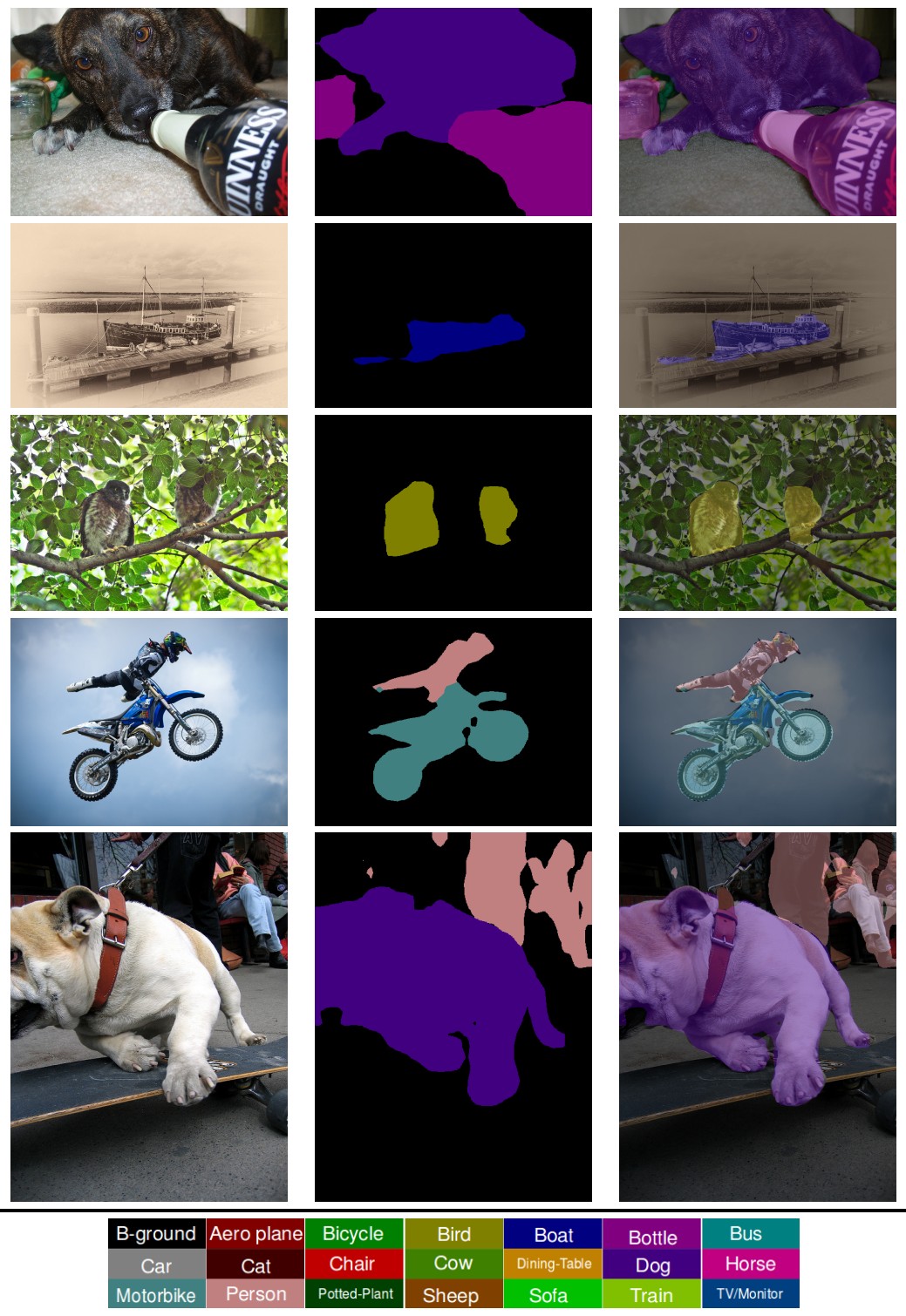

| B-ground | Aero plane | Bicycle | Bird | Boat | Bottle | Bus |
|----------|-----------|---------|------|------|--------|-----|
| Car | Cat | Chair | Cow | Dining-Table | Dog | Horse |
| Motorbike | Person | Potted-Plant | Sheep | Sofa | Train | TV/Monitor |

Figure 18: **Semantic segmentation results** of Deeplabv3-MobileViT-S model on the *unseen* MS-COCO validation set (**left:** input RGB image, **middle:** predicted segmentation mask, and **right:** Segmentation mask overlayed on RGB image). Color encoding for different objects in the PASCAL VOC dataset is shown in the last row.

```python
import torch
from torch.utils.data.sampler import Sampler
import torch.distributed as dist
import math
import random
import numpy as np

class MultiScaleSamplerDDP(Sampler):
    def __init__(self, base_im_w: int, base_im_h: int, base_batch_size: int, n_data_samples: int,
        min_scale_mult: float = 0.5, max_scale_mult: float = 1.5, n_scales: int = 5, is_training: bool =
        False) -> None:
        # min. and max. spatial dimensions
        min_im_w, max_im_w = int(base_im_w * min_scale_mult), int(base_im_w * max_scale_mult)
        min_im_h, max_im_h = int(base_im_h * min_scale_mult), int(base_im_h * max_scale_mult)

        # Get the GPU and node related information
        num_replicas = dist.get_world_size()
        rank = dist.get_rank()

        # adjust the total samples to avoid batch dropping
        num_samples_per_replica = int(math.ceil(n_data_samples * 1.0 / num_replicas))
        total_size = num_samples_per_replica * num_replicas
        img_indices = [idx for idx in range(n_data_samples)]
        img_indices += img_indices[:(total_size - n_data_samples)]
        assert len(img_indices) == total_size

        self.shuffle = False
        if is_training:
            # compute the spatial dimensions and corresponding batch size
            width_dims = list(np.linspace(min_im_w, max_im_w, n_scales))
            height_dims = list(np.linspace(min_im_h, max_im_h, n_scales))
            # ImageNet models down-sample images by a factor of 32.
            # Ensure that width and height dimensions are multiple of 32.
            width_dims = [(w // 32) * 32 for w in width_dims]
            height_dims = [(h // 32) * 32 for h in height_dims]

            img_batch_pairs = list()
            base_elements = base_im_w * base_im_h * base_batch_size
            for (h, w) in zip(height_dims, width_dims):
                batch_size = max(1, (base_elements / (h * w)))
                img_batch_pairs.append((h, w, batch_size))
            self.img_batch_pairs = img_batch_pairs
            self.shuffle = True
        else:
            self.img_batch_pairs = [(base_im_h, base_im_w, base_batch_size)]

        self.img_indices = img_indices
        self.n_samples_per_replica = num_samples_per_replica
        self.epoch = 0
        self.rank = rank
        self.num_replicas = num_replicas

    def __iter__(self):
        if self.shuffle:
            random.seed(self.epoch)
            random.shuffle(self.img_indices)
            random.shuffle(self.img_batch_pairs)
            indices_rank_i = self.img_indices[self.rank:len(self.img_indices):self.num_replicas]
        else:
            indices_rank_i = self.img_indices[self.rank:len(self.img_indices):self.num_replicas]

        start_index = 0
        while start_index < self.n_samples_per_replica:
            curr_h, curr_w, curr_bsz = random.choice(self.img_batch_pairs)

            end_index = min(start_index + curr_bsz, self.n_samples_per_replica)
            batch_ids = indices_rank_i[start_index:end_index]
            n_batch_samples = len(batch_ids)
            if n_batch_samples != curr_bsz:
                batch_ids += indices_rank_i[:(curr_bsz - n_batch_samples)]
            start_index += curr_bsz

            if len(batch_ids) > 0:
                batch = [(curr_h, curr_w, b_id) for b_id in batch_ids]
                yield batch

    def set_epoch(self, epoch: int) -> None:
        self.epoch = epoch
```

Listing 1: PyTorch implementation of multi-scale sampler

