# OpenReview forum: "MobileViT: Light-weight, General-purpose, and Mobile-friendly Vision Transformer"
_ICLR.cc/2022/Conference — ICLR 2022 Poster_

### Official Review · Reviewer_XdK7 · 2021-10-20

**Correctness:** 3
**Technical Novelty And Significance:** 2
**Empirical Novelty And Significance:** 2
**Recommendation:** 5
**Confidence:** 4

**Details Of Ethics Concerns:**

None.

**Main Review:**

# Weaknesses
- How to reflect the low latency attribute of this network?
- Why use unfold operation to get non-overlapping flatten vectors rather than flatten the input feature directly?
- Could the authors provide the comparison of the top5 accuracy metric in Figure-7? Besides, the author should provide more details about the competitors for the readers' convenience. For example, BoTNet > BoTNet-50.
- The reviewer observes that the authors use an exponential moving average strategy during inference. Is the comparison fair for other competitors, such as BoTNet in Figure-7? Note that there is only a 0.1% difference between those two models.
- From Table-3, we can observe that MobileNetV2 (3.5M & 73.3@top1) outperform the performance of DeiT (5.7M & 72.2@top1) and PiT (4.9 & 73.0@top1). This raises the following two concerns: (a) the base model MobileNetV2 can achieve such promising performance. Is the choice of base model contributes to holding the superiority of MobileNetViT? Have you ever tried other base light-weight models? (b) The efficiency-performance ratio of MobileNetV2 is better than the proposed MobileNetViT. Thus, the value in the real-world practice is smaller than the MobileNet families.
- The PASCAL VOC 2012 dataset is too old-fashioned to verify the actual effectiveness of the proposed method. The authors should conduct the experiments on ADE20K [1*].
- Lack of visualisation results in object detection and segmentation tasks.
- Some typos: DeIT > DeiT.

# Reference
[1*] Scene parsing through ade20k dataset. CVPR2017.

**Summary Of The Paper:**

This paper introduces a lightweight and general-purpose vision transformer, termed MobileViT, for mobile devices. It attempts to build a model which can combine the strengths of CNNs and ViTs to build a light-weight and low latency network for mobile vision tasks. MobileViT achieves top-1 accuracy of 78.4% on ImageNet-1k dataset (classification task), 27.7 mAP on MS-COCO dataset (object detection task), and 79.1 mIOU on VOC 2012 dataset (semantic segmentation task).

**Summary Of The Review:**

Overall, I think this paper is well-written and -organised. The promising performance impresses me a lot.

However, this model stack the mobilenet and transformer models, and thus, the novelty and theory may not be enough for the ICLR community's publication. Besides, according to the reviewer's previous experience, the proposed framework can also work well in most lightweight backbones. The reviewer suggests that the authors should conduct extended experiments to verify this framework is a CNN-agnostic strategy. It can provide general insight to the researcher. Finally, this model's mobile device application is limited (though the authors mention this point in the submission).

Overall, the reviewer votes for the borderline in the first round. The reviewer is also happy to change the initial score according to the quality of the rebuttal content.

---

### Official Review · Reviewer_cGXj · 2021-10-31

**Correctness:** 3
**Technical Novelty And Significance:** 3
**Empirical Novelty And Significance:** 3
**Recommendation:** 6
**Confidence:** 4

**Details Of Ethics Concerns:**

No apparent ethics concerns are found.

**Main Review:**

Strength
-	The design of the new MobileViT block, which combines local convolution and global self-attention block, is neat and simple to implement.
-	The paper writing is easy to follow and technical exposition is clear.
-	The experimental results are quite promising.

Weakness
-	If I understand correctly, quite a few factors contribute to the final superior results of MobileViT on three benchmarks, including MobileViT model architecture, multi-scale training, label smoothing, and EMA. The paper claims the model architecture and multi-scale training as the main contributions. Thus, it is important to measure how much label smoothing and EMA improves the final results, and note whether other competing approaches use them during external comparisons.
-	The paper mainly uses number of the parameters to compare model complexity. This is a theoretical metric. I would also like to compare GFLOPS between MobileViT and other competing models (e.g. MobileNet family models, DEIT, MnasNet, PiT).
-	Table 1 (b), comparisons with heavy-weight CNN are not quite convincing. More recent backbones should be considered, such as MobileNetV3, NAS-FPN, DET-NAS.
-	Table 2, PASCAL VOC 2012 is a small benchmark for segmentation. It would be more convincing to include results on more recent bencmarks such as COCO, LVIS. The competing approaches MobileNetv1, v2 and R-101 are not up-to-date. Please consider more recent backbones such as MobileNetV3, NAS-FPN, DET-NAS.
-	Table 2, is there a strong reason to not compare with transformer-based backbone, such as Multi-scale Vision Transformer and Pyramid Vision Transformer.
-	Technical exposition: After Eqn (1), “Another convolutional layer is then used to fuse local and global features in the concatenated tensor”.  This is a bit confusing. According to Figure 1 (b), the concatenation takes the original feature and another feature computed by local conv and global self-attention building block


**Summary Of The Paper:**

This work proposes a hybrid backbone which combines existing MV2 block and new MobileViT block for efficient classification, detection and segmentation. The core idea with MobileViT block is to use both convolution and self-attention block to aggregate local and global feature, respectively. Experiments on IN-1K, COCO and PASCAL VOC have shown quite promising results.

**Summary Of The Review:**

The overall design of MobileViT block is neat, simple and effective. The MobileViT model is a competitive example of combing existing CNN blocks and new transformer-based block to achieve high accuracy and low-latency on mobile CPUs.

The paper quality can be improved by disentangle the improvement from the claimed contributions (i.e. MobileViT architecture, multi-scale training) from the rest (e.g. label smoothing, EMA, and any other training recipe under the hood). This is important for the main results in Figure 6 and 7.

---

### Official Review · Reviewer_GY8P · 2021-11-02

**Correctness:** 3
**Technical Novelty And Significance:** 3
**Empirical Novelty And Significance:** Not applicable
**Recommendation:** 8
**Confidence:** 4

**Main Review:**

- (+) the paper is well written, the contributions are clearly stated and supported by empirical results.
- (+) the proposed backbone architecture exhibits improved performance with respect to previous work based on ViT: robustness to hyperparameters, training stability and generalization capability.
- (+) with the proposed block the authors are able to reduce the number of parameters while preserving or improving performance on classification, object detection and segmentation, suggesting they are indeed providing a good backbone for many downstream tasks.
- (+) the core technical idea of the paper is simple and convincing. I like the idea that by removing the need for positional encoding, authors enable multi-scale training of ViTs.
- (+) the experiments are well designed to support most of the authors' claims. They wisely compare against both famous CNNs and ViT-based recent work. Training and design details are provided and the overall method seems reproducible. The model is trained from scratch, hinting to the optimization robustness claimed by the authors.

The main drawbacks I see in the paper are related to memory and time performance analysis, that should be better clarified given the mobile focus of the work:
- (-) The worst case analysis presented in Sec. 3.1 is not helpful to the reader, since it does not explain eventual speed improvements. The analysis is i) worst case, ii) asymptotic and, to my understanding, iii) the parameters involved change between MobileViT and standard ViTs. I think the analysis should either be fixed or replaced with a FLOPs count comparison.
- (-) Together with the number of parameters and time complexity / FLOPs, memory footprint is another important element for embedded computer vision algorithms based on deep learning. This is particularly true given the notoriously poor performance of transformers on this topic.
- (-) MobileViT is compared against MobileNetV2, DeIT and PiT on an iPhone 12. It would be interesting to see the same comparison on a number of different devices, e.g. a low end mobile device, as well as GPU (for which only training timings are reported) and CPU.

Minor issues (do not contribute to review scoring):
- at the end of Sec. 4.3 Mobile-friendly, there is a reference to "larger inputs" not quantified until later in the appendix. Since the authors are drawing a conclusion here, it would be more appropriate to state it.
- in the abstract and in Sec. 3 the term "transformers as convolutions" is used, but it is only explained later in Sec. 3.1 Relationship to convolutions. At this point the reader cannot make much sense of those words. I would suggest a rephrasing, or dropping the term altogether as I personally don't think it adds anything to the understanding of the contributions... I actually found it confusing.

**Summary Of The Paper:**

In this paper the authors introduce a new block that combines the global receptive field of transformers with the positional bias of convolutions. They build on this block to produce an architecture that compares favourably against both well known CNNs and classical ViT solutions in terms of number of parameters and task performance ratio.

**Summary Of The Review:**

The contribution is interesting and claims are supported. Unfortunately, the analysis part is a bit lacking as I highlighted in the main review section.

---

### Public Comment · ~Xiangxiang_Chu1 · 2021-11-17
**Performance comarison with recent vision transformers**

1. Regarding recent vision transformers [1,2,3] obtain significantly boosted performance (7+mIOU or mAP) on down stream tasks (COCO detection and ADE segmentation) than CNNs.  It would be better that MobileViT is compared with them by controlling similar flops. For example, their codes are released, and it's not difficult to construct the baselines by reducing the number of channels.
2. The latency of MobileViT is too high (about one order of magnitude  slower, compared with Mobilenet V2 and V3), any suggestion to make it applicable in practical applications?

[1] Swin Transformer: Hierarchical Vision Transformer using Shifted Windows, ICCV21,
 https://github.com/microsoft/Swin-Transformer

[2] SegFormer: Simple and Efficient Design for Semantic Segmentation with Transformers, NEURIPS21,
 https://github.com/NVlabs/SegFormer

---

### Decision · Program_Chairs · 2022-01-20

**Decision:**

Accept (Poster)

**Comment:**

This paper presents a light weight hybrid model using both convolutions and Transformer layers resulting in models with lower computational cost and good performance. Reviewers find the paper interesting and agree that the paper did a good job in presenting convincing experimental results. There were questions about role of different components in the proposed model, which author's addressed in the response with additional ablation studies. One reviewer expressed concerns about lack of theoretical foundations for the proposed approach. However they also agree that the paper presents a good and useful experimental study. I think overall the paper has good contributions that others can build on in the future and recommend acceptance.